# Surgical Management of Retroperitoneal Sarcoma

Dana A. Dominguez [1] , Sagus Sampath [2], Mark Agulnik [3] , Yu Liang [4], Bao Nguyen [5], Vijay Trisal [1], Laleh G. Melstrom [1], Aaron G. Lewis [1], Isaac Benjamin Paz [1], Randall F. Roberts [6] and William W. Tseng [1],*

1    Division of Surgical Oncology, Department of Surgery, City of Hope National Medical Center, 1500 East Duarte Road, Duarte, CA 91010, USA; dadominguez@coh.org (D.A.D.)
2    Department of Radiation Oncology, City of Hope National Medical Center, 1500 East Duarte Road, Duarte, CA 91010, USA
3    Department of Medical Oncology and Therapeutics Research, City of Hope National Medical Center, 1500 East Duarte Road, Duarte, CA 91010, USA
4    Department of Pathology, City of Hope National Medical Center, 1500 East Duarte Road, Duarte, CA 91010, USA
5    Department of Diagnostic Radiology, City of Hope National Medical Center, 1500 East Duarte Road, Duarte, CA 91010, USA
6    Division of Thoracic Surgery (Vascular Surgery Section), City of Hope National Medical Center, 1500 East Duarte Road, Duarte, CA 91010, USA
\*    Correspondence: wtseng@coh.org; Tel.: +1-626-218-1414

**Abstract:** Surgery is the cornerstone of treatment for retroperitoneal sarcoma (RPS). Surgery should be performed by a surgical oncologist with sub-specialization in this disease and in the context of a multidisciplinary team of sarcoma specialists. For primary RPS, the goal of surgery is to achieve the complete en bloc resection of the tumor along with involved organs and structures to maximize the clearance of the disease. The extent of resection also needs to consider the risk of complications. Unfortunately, the overarching challenge in primary RPS treatment is that even with optimal surgery, tumor recurrence occurs frequently. The pattern of recurrence after surgery (e.g., local versus distant) is strongly associated with the specific histologic type of RPS. Radiation and systemic therapy may improve outcomes in RPS and there is emerging data studying the benefit of non-surgical treatments in primary disease. Topics in need of further investigation include criteria for unresectability and management of locally recurrent disease. Moving forward, global collaboration among RPS specialists will be key for continuing to advance our understanding of this disease and find more effective treatments.

**Keywords:** retroperitoneal sarcoma; liposarcoma; leiomyosarcoma; multidisciplinary; collaboration

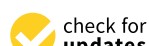



## 1. Introduction

Soft tissue sarcomas encompass a group of rare, histologically diverse malignancies with distinct genetic aberrations and clinical behavior. Fifteen to 20% of soft tissue sarcomas can occur in the back of the abdomen or retroperitoneum. In this anatomic location, the tumors are frequently massive and in fact, retroperitoneal sarcomas (RPS) are among the largest tumors in the human body. In addition to size, the tumors can involve critical organs and structures, making surgery challenging. Importantly, the histologic types of RPS are limited to liposarcoma (the most common–Figure 1), leiomyosarcoma (Figure 2) and others such as solitary fibrous tumors, undifferentiated pleomorphic sarcoma and malignant peripheral nerve sheath tumors. Recognition of the precise histologic type is critical to treatment decision-making in RPS. While surgery is the cornerstone of treatment, there are important details and nuances beyond "cutting it out" that will be discussed in this review. Unique to this review, we present a practical approach to the surgical management of RPS that addresses controversies with available evidence, from small studies to landmark trials, and continuously recognizes the importance of disease biology in decision-making.

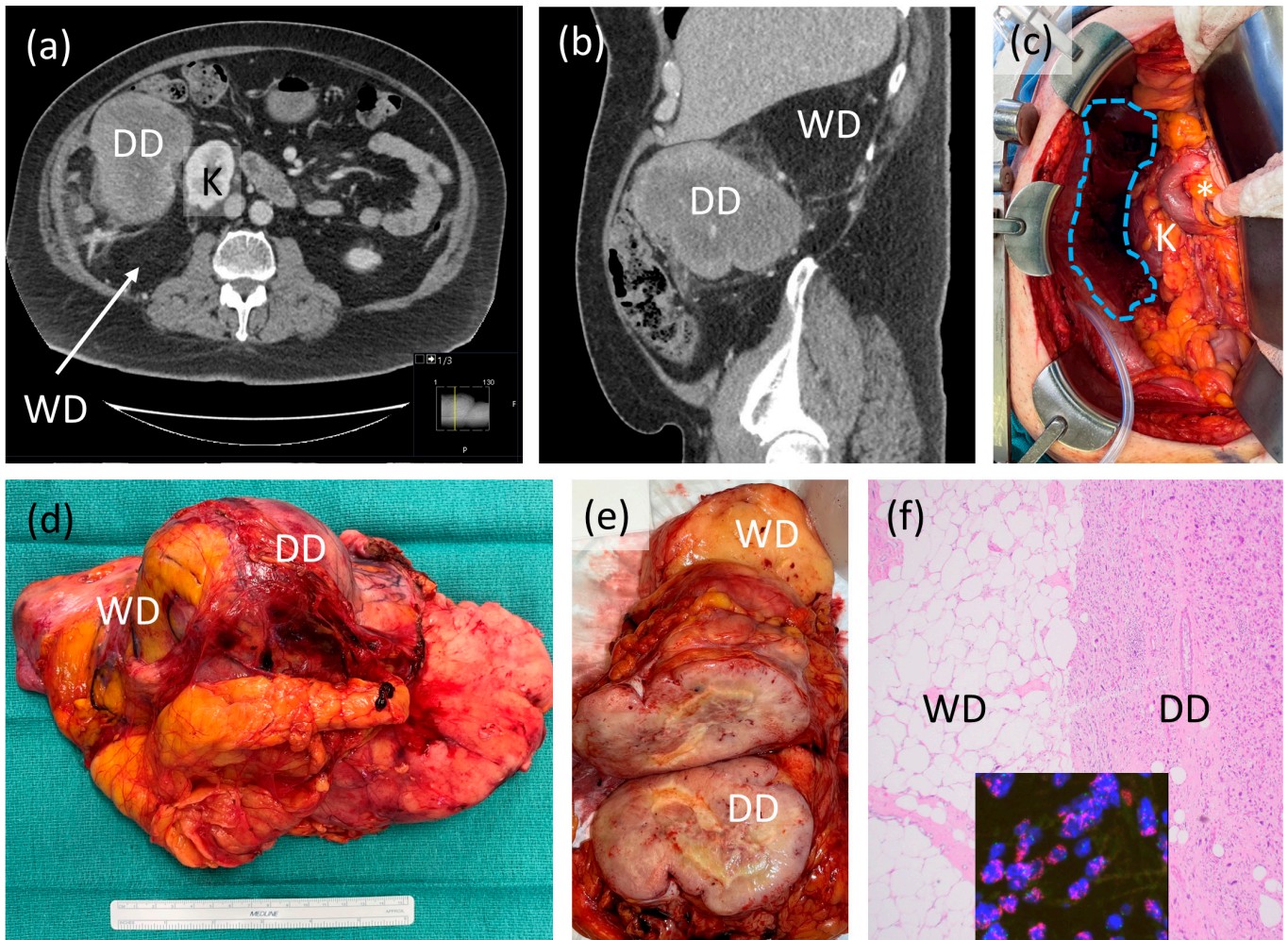

**Figure 1.** Case example of retroperitoneal liposarcoma. (**a**,**b**) Axial and sagittal CT scan images showing likely high-grade or dedifferentiated (DD) and low-grade or well differentiated (WD) components of the tumor, abutting the right kidney (K); (**c**) intraoperative photograph after resection showing original area of tumor (blue dotted shape) with preserved right kidney (K), duodenum and head of pancreas (*). Organ-preservation is considered on an individual case basis; (**d**) gross resection specimen demonstrating one intact tumor with WD, DD components and some incorporated, normal-appearing adjacent fat; (**e**) sectioned gross tumor showing in this case, clear demarcation of WD and DD; (**f**) photomicrograph (40×) of tumor histology showing clear demarcation of WD and DD, inset showing MDM2 amplification by fluorescence in situ hybridization, confirming the diagnosis. This patient did not receive neoadjuvant or adjuvant therapy.

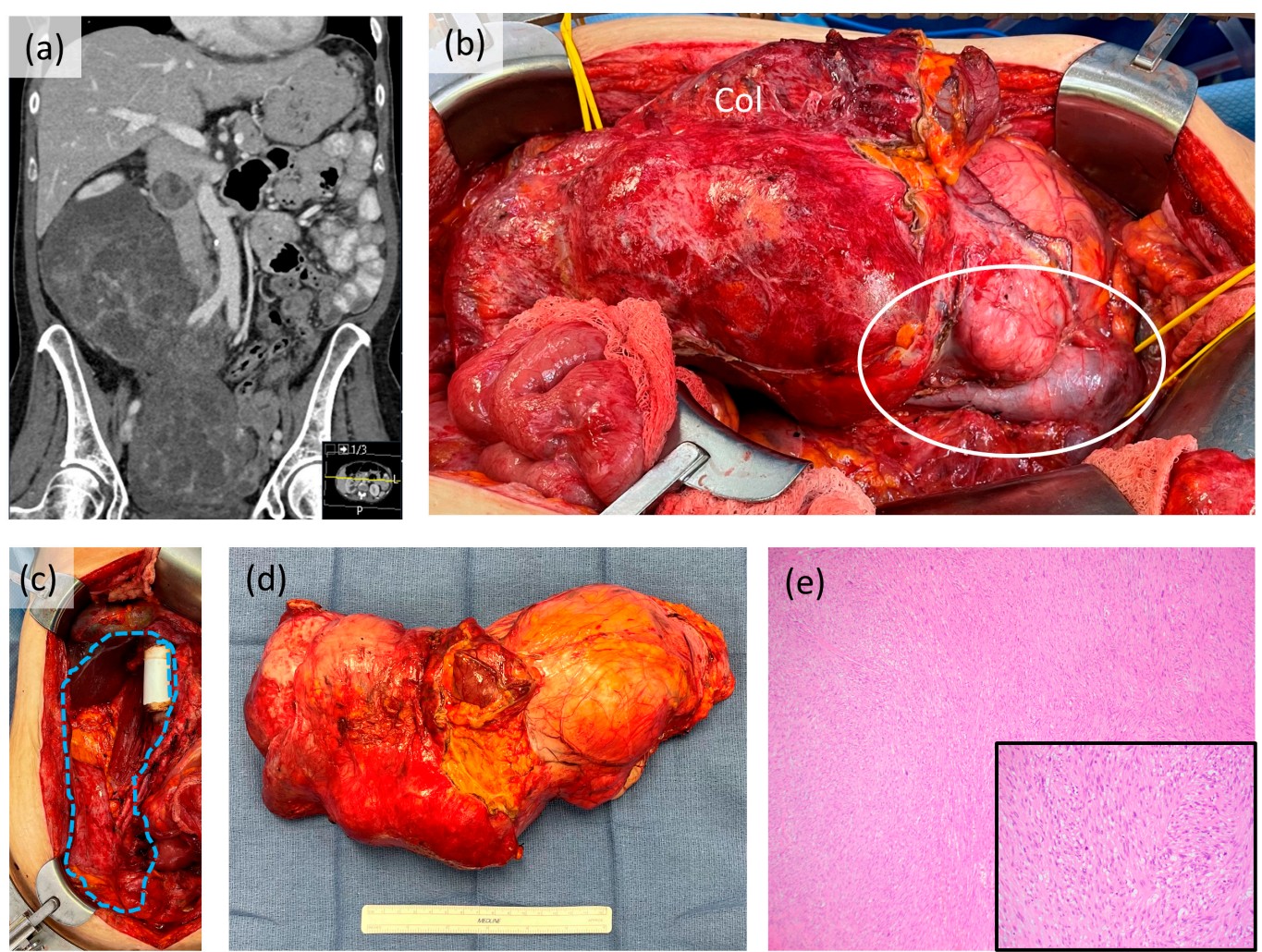

**Figure 2.** Case example of retroperitoneal leiomyosarcoma. (**a**) Coronal CT scan image showing large tumor extending into the pelvis with intraluminal tumor extension in the inferior vena cava; (**b**) intraoperative photograph prior to resection showing gross involvement of the right colon (Col) and intracaval tumor component (white circle). The right kidney was also involved; (**c**) intraoperative photograph after resection showing original area of tumor (blue dotted shape) with in-line IVC graft reconstruction performed by vascular surgery; (**d**) gross resection specimen; (**e**) photomicrograph (40×) of tumor histology, inset (200×). This patient received neoadjuvant systemic therapy with good response radiologically, clinically and pathologically.

## 2. Initial Work-Up in Primary Disease

Patients with RPS can present subtly with vague and non-specific symptoms. Some patients can be entirely asymptomatic with their tumor discovered incidentally on imaging (e.g., ultrasound or MRI) carried out for another purpose. If not previously carried out, the work-up of suspected RPS should include a good quality, contrast-enhanced CT of the abdomen and pelvis. This is critical for surgical planning purposes. In some cases, MRI can add additional information to better delineate soft tissue involvement [1].

### 2.1. Needle Biopsy

The imaging features of the retroperitoneal tumor can be used to predict the histologic type but ultimately, the definitive diagnosis requires tissue sampling. Core needle biopsy before surgery, while not required in every patient, may potentially provide useful information if performed. A biopsy can rule out benign entities that can mimic RPS (e.g., angiomyolipoma and others) [2], confirm the RPS diagnosis and, importantly, establish the

histologic type. This can be useful to counsel the patient and guide subsequent histology-specific management. If neoadjuvant therapy is being considered, obtaining this pathologic information upfront is critical. In general, while a needle biopsy can also provide the tumor grade, the accuracy of this assessment is controversial. Ultimately, detailed pathologic evaluation is best carried out on the resection specimen, accounting for neoadjuvant therapy, if given.

If performed, needle biopsy is carried out by an interventional radiologist through image (e.g., CT)-guidance to target a non-necrotic, solid, avidly enhancing portion of the tumor. The biopsy is carried out with a retroperitoneal approach, avoiding the free abdominal cavity, using a co-axial technique with at least four passes using an 18-gauge needle to maximize tissue available for the pathologist [3]. When performed appropriately, the frequency of complications and biopsy tract seeding is extremely low to nil in RPS [4,5]. Surgical incisional biopsy, whether open or performed in a minimally invasive manner, is discouraged as this can disrupt tissue planes for subsequent resection and risk tumor seeding.

### 2.2. Staging

Once the diagnosis is established, for RPS histologic types with metastatic potential, at minimum, a CT of the chest should be obtained for complete staging. The lungs are the most frequent site of metastasis, and the detection of distant disease would obviate plans for surgery. When obtaining cross-sectional imaging for staging, potential sites of metastasis for specific RPS histologic types (e.g., liver in leiomyosarcoma) should be recognized. None of the RPS histologic types involve the regional lymph nodes. A whole-body PET scan is not standard for staging; however, in RPS, this imaging modality is being further investigated as a prognostic tool for the primary tumor [6–9].

### 2.3. Importance of Management by a Sarcoma Specialist

For suspected or diagnosed RPS, it is critical for the patient to be initially evaluated and ultimately managed by a surgeon with expertise in this disease. The surgeon is typically a fellowship-trained surgical oncologist with experience and up-to-date knowledge of sarcoma, in some cases even focusing their clinical practice entirely on this disease [10]. The surgical oncologist should be an active, integrated member of a multidisciplinary team of sarcoma specialists that includes medical- and radiation oncologists, as well as pathologists and radiologists. When RPS patients are seen by specialists, the outcomes are markedly improved [11–14], which would be intuitive for an ultra-rare, complex malignancy. A volume of 10–13 RPS cases per year has been cited as the minimum threshold for specialist designation [11,13], although this is controversial and should likely include more than just number of cases [10,15].

## 3. Surgery

Surgery is the cornerstone of treatment in RPS. For primary disease, this is frequently carried out upfront once the diagnosis is established and staging has been completed, verifying the absence of metastatic disease. RPS surgery is challenging given the frequently large size of the tumor and close proximity to critical organs and structures within the limited space of the retroperitoneum and intraabdominal cavity.

### 3.1. Preoperative Planning

The surgeries performed for RPS are major undertakings that involve planning and preparation, even before arriving at the operating room. RPS operations often last many hours, can involve the concurrent resection of multiple organs/structures and, overall, may be associated with significant blood loss, risk of complications and even intra- or postoperative death. The RPS patient, often anxious to "get it out", should be counseled appropriately about the potential extent of surgery and subsequent impact on quality of

life. While preoperative imaging is useful to anticipate important details of resection, the true extent of surgery may only be realized intraoperatively.

Any pre-existing co-morbidities should undergo medical evaluation in anticipation of major surgery. Those with borderline or pre-existing renal insufficiency may need further investigation (e.g., split renal perfusion scan) to determine the tolerance of an ipsilateral nephrectomy. Those with significant cardiopulmonary disease should undergo updated evaluation (e.g., echocardiogram) and, if needed, intervention to improve organ function as much as possible. Patients with diminished reserve may not tolerate a major RPS operation and data from preoperative medical evaluation can impact decision-making for RPS surgery. Importantly, for all RPS patients, careful screening for malnutrition should be carried out and protein-caloric intake improved to help optimize the outcome for surgery [16–18].

Depending on individual and institutional practice, the surgeon may find it useful to obtain outpatient consultations with other surgical services as part of the preoperative planning for RPS surgery. This may include the vascular surgeon if major vessel resection and reconstruction are likely, or the urologist, for intraoperative stent placement or possible extended bladder resection. As RPS surgery can go beyond the typical "abdominal case", advanced discussion with the anesthesiologist is often helpful to anticipate potential intraoperative needs (e.g., blood products) and the appropriate level of venous access and hemodynamic monitoring.

### 3.2. Surgical Approach

The standard incision most commonly used is a wide midline laparotomy, as this permits the most access to the tumor and the great vessels (aorta, inferior vena cava or IVC, and iliac vessels). Lateral extensions on the side of the tumor may be added or an entirely different incision (e.g., lateral thoracoabdominal) may also be appropriate, depending on each case and the surgeon preference. The key to any incision in RPS surgery is that it needs to provide optimal, safe exposure to the tumor and surrounding organs and critical structures.

The goal of RPS surgery is to achieve the complete en bloc resection of the tumor along with involved adjacent organs and structures to maximize clearance of the disease. The type of resection can be defined as R0 (complete resection with negative microscopic margins), R1 (complete resection with positive microscopic margins) or R2 (incomplete resection). In primary disease, the piecemeal, partial removal (debulking/R2 resection) or rupture of the tumor should be strongly avoided, as this has been demonstrated to have a detrimental effect on oncologic outcomes. Multiple large cohorts examining prognostic factors for OS have demonstrated significant hazard ratios from 1.70–2.20 for an R2 resection when controlling for other predictive factors [19–21]. In fact, in some studies, patients who had an R2 resection had equivalent outcomes to patients who underwent biopsy alone, adding the risk of surgical morbidity, without any added benefit [22–24]. Regrettably, inadvertent incomplete resection is sometimes carried out by a non-specialist surgeon for liposarcoma when the obvious high grade or dedifferentiated component of the tumor is removed while the surrounding lipomatous low grade or well differentiated disease is left behind. A non-specialist may also perform lymph node dissection, which is completely unnecessary in RPS and can expose the patient to undue risk (e.g., chylous leak).

At minimum, complete gross resection should be achieved in each case. Microscopic negative margins (R0) can be strived for; however, this is frequently difficult if not overtly impossible to achieve [25]. Within the restricted abdominal space, the large size of RPS tumors would hypothetically necessitate the removal of all abutted organs, structures and even surfaces (e.g., posterior retroperitoneum). Accurate pathologic margin assessment of all "inked" surfaces on the resection specimen is also not practical, particularly for larger tumors. If R0 resection is truly achieved, this is likely associated with an improved outcome [26]; however, the data to support this is likely biased (e.g., smaller tumors). The consensus among most sarcoma specialists is that in describing RPS surgery, it is more appropriate to distinguish between R2 (gross disease left behind) and R0/R1 resection.

The obvious tumor involvement of organs and structures, ideally anticipated during preoperative planning (e.g., imaging review), does necessitate concomitant resection. In recent, large series reported by single and multi-institutional sarcoma centers, resection of one or more organs is carried out in 58–87% of all cases of primary RPS [27,28]. Most commonly, ipsilateral nephrectomy and partial colectomy are performed in more than half (55–57%) of all cases. With appropriate planning and available support (e.g., vascular surgeon), resection of involved major vessels (e.g., IVC) can be performed; however, the frequency of this is less common (10–15%). An additional, laterality-specific organ resection may be needed depending on the patient case. For left sided tumors, distal pancreatectomy and splenectomy may be required [29]; for right sided tumors, pancreaticoduodenectomy (Whipple) can be considered, but is in fact rarely carried out (1.4% of cases) [30].

To maximize the clearance of the disease, sarcoma specialists in Italy and France have described an extended or compartmental approach to resection [21,31]. The fundamental concept is that even without obvious tumor involvement, adjacent organs, structures and even surfaces (e.g., psoas fascia) should be resected en bloc with the tumor in an effort to obtain circumferential "soft tissue margins". This is analogous to obtaining wide soft tissue margins in high grade, extremity sarcoma but adapted to the retroperitoneum. When first introduced, this approach generated controversy [32,33] and to date, extended or compartmental resection in primary RPS is not universally accepted across sarcoma centers, particularly in the United States. In concept, in an appropriately selected patient, when technically feasible and safe, extended resection should be applied to strive for an R0 resection; in daily practice, extended resection may have a more limited application to liposarcoma and after consideration of the entirety of each patient's situation (e.g., co-morbidities, expected oncologic outcome).

For organ abutment without obvious tumor involvement, the decision to resect should consider the "expendability" of the organ and risk for complications. One kidney or a portion of colon is well tolerated in most patients. By comparison, in a recent review of major vascular resection RPS, sarcoma specialists have advocated for the dissection of tumor-abutted vessels if feasible, as opposed to resection [34]. Aligned with this, even for extended or compartmental resection, component procedures that are potentially morbid (e.g., major vascular resection or Whipple) are carried out only for overt invasion [35].

Several studies have investigated the frequency of microscopic infiltration in resected organs in an attempt to help guide surgical decision making [36,37]. These data are inherently biased in that only resected tissue is available for study and the level of pathologic assessment is not standardized. Histologic organ invasion is common, but not universal across all resected organs and the frequencies of involvement vary by organ, the histologic type of RPS and from study to study. As such, the decision to resect abutted organs and structures is arguably still controversial and, clinically, one must consider potential morbidity while maintaining oncologic principles (e.g., no gross violation of tumor integrity) with the goal of at least a complete en bloc resection.

A histology-based approach to RPS surgery in primary disease has recently gained increased recognition among sarcoma specialists [7,25,38,39]. This surgical approach considers the anticipated origin and local extent of the tumor based on an intimate understanding of histologic type. For example, in the retroperitoneum, a (right sided) leiomyosarcoma may arise from the IVC (Figure 2) and a malignant peripheral nerve sheath tumor from the nerve root. As such, the maximum clearance of disease is focused on these key areas while achieving minimum complete gross resection elsewhere. In fact, frozen section analysis of these key "margins" (e.g., IVC in leiomyosarcoma) may be beneficial. For liposarcoma, the need to further clear "at risk" adjacent fat remains controversial and may support an extended resection. A histology-based approach in retroperitoneal liposarcoma must incorporate differences (e.g., local invasiveness) in tumors that are entirely well differentiated versus those with a dedifferentiated component. A lipoma-like well differentiated liposarcoma next to the duodenum and head of pancreas, as an example, is very different from a grade 3 dedifferentiated liposarcoma, even when controlled for tumor size. For all

retroperitoneal liposarcoma, there is also the possibility of multifocal disease–defined as the synchronous presence of two or more tumors–which although rare, can occur in primary disease [40,41]. Importantly, with histology based RPS surgery, the surgical approach (e.g., need for extended resection) is also dependent on the potential patterns of future recurrence. Extended resection may not be necessary for leiomyosarcoma which has a much greater risk of distant than local recurrence or for a solitary fibrous tumor which has a minimal risk for either event [39].

Beyond primary disease, the surgical management of locally recurrent RPS is less well-defined. In this patient population, negative prognostic factors include a higher number of organs resected at the initial surgery, age at second surgery, multifocality at second surgery, high grade, incomplete resection and dedifferentiated liposarcoma histology [42]. Local recurrence is an especially challenging issue for retroperitoneal liposarcoma. In this histologic type, data from a single high volume sarcoma center suggests that a growth rate of greater than one centimeter per month can be considered to identify patients that may not benefit from resection [43]; however, this "rule" is not universally accepted. In liposarcoma, the issue of local recurrence is further complicated by second, even third and beyond recurrences, as well as late recurrence after a prolonged disease-free interval. While there exists some skepticism about the benefit of surgery in locally recurrent RPS [44], there is also data to support surgery in select patients, even after multiple recurrences [45,46]. The decision to operate on recurrent disease is complex and in daily practice should be discussed among a multidisciplinary team of RPS specialists and consider other treatment modalities.

### 3.3. Morbidity of Surgery

The operations performed for RPS are often challenging and overall, the morbidity can be substantial. In a large multi-institutional series of patients with primary disease, the frequency of severe complications, defined as Clavien-Dindo grade 3 or higher, was 16.4% [47]. Reoperation was required in 10.5% of patients and within 30 days of surgery, 1.8% died. Not surprisingly, resections that involved major vessels or pancreaticoduodenectomy had the highest association with severe complications. In this series, the frequency of any grade complication was not reported. In other single and multi-institutional series from sarcoma specialist centers, these data range from 27–34% with a recent outlier of 82.9% [48–50].

As discussed, concomitant multiorgan resection occurs often in RPS surgery and the risk of complications can be additive with each component of the operation. As an example, an RPS patient who undergoes concomitant distal pancreatectomy/splenectomy or left colectomy may develop a leak that could jeopardize an iliac artery reconstruction. Clinically, the totality of complications can also be subtle (e.g., additional massive third-spacing in a post-nephrectomy patient) and compounded by pre-existing co-morbidities and malnutrition, when present. As a result, there is increasing recognition of the need for better tools to account for these situations, such as the "comprehensive complication index" [50–52]. The uniqueness and complexity of RPS surgery again highlights the need for management by a specialist.

### 3.4. Outcomes after Surgery

Despite aggressive surgery with en bloc multiorgan resection, LR in primary RPS remains high. Reported LR rates from specialist centers vary up to 49% at 5 years [21,27,28,53–57]. These rates were not significantly improved, even in cohorts reporting the highest rates of complete resection (>90%) [27,53]. Multiple studies have identified factors associated with increased LR risk, including histologic type, grade, receipt of radiation therapy and completeness of resection [25]. Extended or compartmental resection with the liberalized en bloc resection of adjacent organs/structures is associated with lower reported rates of LR [21,31]. In the single center study from Italy, a significant decrease in LR (49.3 vs. 27.8%, $p < 0.0001$) was found when comparing patient outcomes before and after the implementation of this surgical approach [31]. Similarly, in a multi-institutional study from

France, patients who underwent extended resection had a 3.3-fold decrease in LR versus those who underwent standard (but complete) resection [21].

In contrast to LR, rates of distant metastasis (DM) in primary RPS have remained fairly constant, ranging from 12–22% in the largest cohorts [25]. The histologic type is the key distinctive factor determining the pattern of recurrence, specifically the risk for LR versus DM. In a large multi-center study of 1007 RPS patients, after complete resection by a specialist, those with well-differentiated liposarcoma had an 8-year LR rate of 22%, without any patient with DM. By comparison, patients with leiomyosarcoma had an 8-year LR rate of 10% but had the highest rate of DM among all histologic types: 50% at 8 years [25,27]. Patients with dedifferentiated liposarcoma had outcomes "in the middle" with LR rates of 36–43% and DM rates of 8–31% [25,27].

Over the last several decades, reported data show improved overall survival (OS) in patients with primary RPS. As an example, in the 1990s, the Dutch Sarcoma Group reported a 5-year OS rate of 39% [56], whereas more recent single and multi-institutional cohorts have reported 5-year OS rates of 67–69% [27,28]. One recent study specifically examined trends in OS over a 15-year period among 10 sarcoma centers [58]. When three periods of time were compared, the authors found an improvement in OS from 61.2% (earliest) to 71.9% (most recent). The study also noted significantly lower 90-day postoperative mortality over the time periods, concluding that the improved survival for patients undergoing resection for primary RPS was likely due to better patient selection, quality of surgery and better perioperative management [58].

## 4. Non-Surgical Therapies

### 4.1. Radiation Therapy

The role of radiotherapy (RT) as part of curative-intent management for primary localized RPS remains a highly debated topic. Given the well-established local-control benefit for adjuvant RT in extremity sarcoma, there has been strong motivation over many decades to examine where a similar benefit exists for RPS. Multiple retrospective series in RPS have reported a local control benefit with RT, although such results have been often questioned due to selection bias [31,59,60]. Unfortunately, accrual to randomized trials has been nothing short of challenging, which is confirmed by the premature closure of the ACSOG Z9031 trial in 2014 [61]. Our highest level of evidence thus far comes from the recently reported phase III STRASS trial evaluating radiotherapy in the neoadjuvant setting [49]. The primary objective of the study was abdominal recurrence-free survival (ARFS), a composite endpoint consisting of multiple factors, including the progression of disease during RT precluding resection. The top line results of the trial were negative, with no difference in ARFS between the surgery and preoperative RT arm compared to the surgery alone arm. The RT dose was 50.4 Gy in 28 fractions. Based on these results, it was concluded that RT does not provide a meaningful benefit for these patients [49].

Advocates of RT for RPS point out several key findings in the trial which are notable. First, RT was associated with a >50% reduction in local relapse in all patients. Second, on the central quality review of RT plans, approximately 25% of plans were found to have major deviations, which would be considered unacceptably high. In fact, 65% of the protocol deviations were related to inadequate gross tumor volume delineation during treatment planning, which may have dampened the results on the RT arm [62]. Third, the trial did not robustly stratify patients by histology, as we have gained significantly more knowledge on clinical behavior of sarcoma types since the time the trial was being originally designed. On a post-hoc exploratory sensitivity analysis, patients with liposarcoma were found to have improved ARFS that achieved borderline statistical significance (HR 0.62, 95% CI 0.38–1.02). Further post-hoc subgroup analysis indicated that benefit to be associated with well-differentiated histology, not higher-grade entities.

Finally, a recent study performed a propensity-matched analysis of more than 1000 patients, combining those treated on STRASS, and those treated off protocol known as STREXIT [63]. The investigators found significantly improved ARFS with pre-operative RT

in patients with grade 1 and grade 1–2 dedifferentiated liposarcoma, but not in leiomyosarcoma or grade 3 liposarcoma. In conclusion, the role of RT in RPS remains in need of further refinement, as we observe outcome differences based on histology. Future prospective trials with RT should incorporate histology-specific stratification when achievable to confirm the STRASS and STREXIT findings [63].

### 4.2. Systemic Therapy

The role of chemotherapy used either in the neoadjuvant, or adjuvant setting, or concurrent with radiation therapy has established roles in the treatment of soft tissue sarcomas, but the specific role in RPS is less established. Given this uncertainty, it is suggested that patients participate in ongoing clinical trials, where available.

Neoadjuvant chemotherapy for RPS patients with resectable disease is not a standard approach but can be considered in those with at least an intermediate grade tumor and whose histology is sensitive to the effect of chemotherapy. When extrapolated from data in extremity soft tissue sarcoma, chemosensitive histologic types include synovial sarcoma, myxoid round cell liposarcoma and angiosarcoma [64]; however, these are rarely found in the retroperitoneum. Until more data is available, chemotherapy in RPS should be reserved mostly for disease that is unresectable or metastatic.

In select cases, neoadjuvant chemotherapy can be considered in RPS patients with borderline resectable tumors for the intent of decreasing tumor size to facilitate surgery. In a recent multi-institutional retrospective study, 23% of patients with RPS had RECIST partial response (>30% decrease in tumor size) to neoadjuvant chemotherapy [65]. The study was not designed, however, to assess whether response improved resectability. All patients underwent complete resection. Patients with disease progression (21%) to chemotherapy before surgery had markedly worse survival, raising the question of whether response to therapy can be used to select patients who may not benefit from proceeding with resection.

STRASS2, a randomized phase III study of neoadjuvant chemotherapy followed by surgery versus surgery alone for patients with high-risk retroperitoneal sarcoma is currently accruing patients globally. This study is specifically evaluating patients with high risk leiomyosarcoma or liposarcoma. All patients will be planned for curative-intent surgery within 4 weeks following randomization. Patients in the experimental arm will receive three cycles of neoadjuvant chemotherapy starting within 2 weeks following randomization followed by curative-intent surgery within 3–6 weeks of the last cycle of chemotherapy. The primary endpoint is disease-free survival. This study will be powered to establish the role of neoadjuvant care for these patients [66].

Adjuvant chemotherapy is somewhat controversial for soft tissue sarcoma at any site, with some retrospective data suggesting improved overall survival with an adriamycin and ifosfamide combination, but this would not be in a patient population enriched with RPS [67]. As such, until more data is available, adjuvant chemotherapy should be considered carefully on an individual basis.

A role for the concurrent administration of preoperative radiation and chemotherapy (chemoradiation) in patients with RPS is not established, and as such, this approach is experimental. An effort to establish whether preoperative chemoradiation is better than preoperative radiation alone would need to be carried out in a prospective way.

## 5. Conclusions and Future Directions

Surgery is the cornerstone of treatment for RPS; however, the management of this histologically diverse group of tumors is complex and nuanced. This must be recognized by the surgeon, who ideally is a specialist in RPS, and incorporated with the multidisciplinary decision-making that goes beyond just "cutting it out".

The surgical approach must strive to achieve the goal of complete en bloc resection, maximizing clearance of disease, while balancing morbidity and an intimate understanding of the anticipated outcomes after surgery for the histologic type of RPS. Specific surgical considerations were discussed above, and further technical details have also been described

elsewhere [35]. The management of primary RPS is rapidly evolving. With the STRASS study completed and STRASS2 open and enrolling patients, neoadjuvant radiation and systemic therapy, respectively, are being more rigorously assessed for their benefit as non-surgical treatments in RPS. This review summarizes the surgical management of RPS. Consensus guidelines for management of primary RPS are also available and have been recently updated [68].

Understanding that each RPS case is unique, and that surgical skill may vary between surgeons, one topic in need of further investigation and, ideally, standardization in the field is criteria for unresectability. Technical points of unresectability may include overt tumor involvement of the superior mesenteric vessels or invasion into the spine; but these have not been universally agreed upon. Importantly, at least equal weight should be given to patient co-morbidities and disease biology, including the histologic RPS type. A single-institution study from a high volume RPS center reported a 12% frequency of unresectability [69]; to date this is the only published study focused on this topic.

While the vast majority of studies in RPS management focus on primary disease, in daily practice, there is a substantial and growing population of patients with recurrent disease, especially those with liposarcoma, who are being evaluated for surgery. In this review, we only briefly mention this topic. Locally recurrent RPS arguably requires even more complex, histology- and disease biology-driven decision making than primary disease and represents another major topic for focused investigation [41]. Until further studies are completed, recent guidelines for management of recurrent RPS can serve as a framework for the care of these patients [70].

Moving forward, more high-quality research is critically needed to continue to advance our understanding of RPS and find more effective treatments [3]. In addition to clinical studies, basic and translational research focused on the biology of RPS (e.g., molecular mechanisms) is key. Global collaboration among specialists is also important to accelerate any research effort as well as to validate and apply relevant findings back to RPS patients. This perspective is at the core of the Transatlantic Australasian Retroperitoneal Sarcoma Working Group (TARPSWG). This international group, founded in 2013, has sponsored many of the major retrospective and prospective (STRASS, STRASS2) studies in RPS and continues to leverage the power of multicenter collaboration to advance the field for this rare and challenging disease [71,72].

**Author Contributions:** Conceptualization, W.W.T.; methodology and data curation (literature search), W.W.T., D.A.D., S.S. and M.A.; writing—original draft preparation, W.W.T.; writing—review and editing, D.A.D., S.S., M.A., B.N., Y.L., V.T., L.G.M., A.G.L., I.B.P. and R.F.R.; supervision and project administration, W.W.T.; funding acquisition, W.W.T. All authors have read and agreed to the published version of the manuscript.

**Funding:** William Tseng's research is partially supported by a grant from the Leiomyosarcoma Support and Direct Research Foundation.

**Conflicts of Interest:** The authors declare no conflict of interest.

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
