# Peer review of "Surgical Management of Retroperitoneal Sarcoma"

_curroncol, doi:10.3390/curroncol30050349_

Round 1
Reviewer 1 Report
Through fully sorted out references, the authors systematically evaluated every step of the surgical management of retroperitoneal sarcoma. "Based on the MDT platform composed of experts from relative disciplines specializing in sarcoma, guided by histological types, and depending on the feasible, safe, and controllable surgical techniques, to ensure complete resection (R0/R1) and strive for R0 resection" is the principle we currently follow in retroperitoneal sarcoma surgery. However, for this kind of tumor with low incidence, diverse pathological types and high heterogeneity, retroperitoneal sarcoma, there is no high-level evidence to resolve surgical issue, and consensus can only be reached on some specific clinical issues.
Regarding the assessment of resectability, in addition to the characteristics of the tumor itself, the surgical team's surgical ability is also a very important factor. So in different centers, facing the same case, different conclusions may be drawn. The same problem is faced with the evaluation of surgical margins, where technical bottlenecks limit the evaluation of margins. For pathologists, in the face of such a large tumor, the evaluation of the circumferential margin is almost technically impossible. Recently, AI technology based on radiographic and pathological images may be helpful in the assessment of tumor resectability and postoperative margin status. In addition, in vivo imaging based on infrared imaging may also be useful in the evaluation of tumor margins during surgery.
Thirteen cases per year are statistically determined to be the cut-off for high-volum center of retroperitoneal sarcoma, but it is very controversial. Referring to the clinical study of laparoscopic pancreaticoduodenectomyâ‘´ , it is not difficult to find that the number of surgical cases has a decisive impact on the surgical outcome. Thirteen cases per year, which is too few for retroperitoneal sarcoma.
Although core needle biopsy can help to preoperative diagnosis, in the clinical practice, it is almost difficult to obtain sufficient and detailed clinical pathological evaluation . thus It is not likely to achieve accurate and individualized clinical decision-making based on histology and grade.
Above all suggest that retroperitoneal sarcoma surgery is an emerging discipline that needs to face many challenges. Mutual reinforce of high volum sarcoma center and high-quality MDT team can is the basis for solving various clinical problems of retroperitoneal sarcoma. Promoting international communication and cooperation, and conducting high-quality clinical study are the goals of TARPSWG. High quality clinical studies, the translations of cutting-edge technology and the breakthroughs in the molecular mechanism of sarcoma will be conducive to the development of retroperitoneal sarcoma surgery.
Reference
1. Wang M, Peng B, Liu J, et al. Practice Patterns and Perioperative Outcomes of Laparoscopic Pancreaticoduodenectomy in China: A Retrospective Multicenter Analysis of 1029 Patients. Ann Surg. 2021;273(1):145-153.
Author Response
We thank the reviewer for their comments on our review. We do agree that there is varying skill between surgical teams, we have added a sentence to mention this fact on line 402. Additionally, we recognize the promise of emerging technologies such as AI and radiomics, however, feel that given we aim to provide a practical approach to RPS management in this review, we feel that mention of these is outside the current scope of this manuscript.
We agree that 13 cases is likely insufficient to be considered a “high-volume” sarcoma center, however, this is existing evidence for a suggested cutoff and felt that was important to highlight, which is why we did include that it remains a controversial number.
We recognize that the use of needle biopsies remains controversial, therefore, we sought to present the data for potential risks and benefits without taking a definitive stance on the topic. We added some text to lines 75-91 to further clarify this.
We agree that MDT, international collaboration, and more high level studies are critical pieces in RPS management, and have added further language in the conclusion section on lines 423-428 to highlight this fact.
Reviewer 2 Report
Section 2.1 line 71 - would suggest that biopsy should be performed in all cases prior to definitive resection
Author Response
We thank the reviewer for their comment. We recognize that the use of needle biopsies remains controversial, therefore, we sought to present the data for potential risks and benefits without taking a definitive stance on the topic. We added some text to lines 75-87 to further clarify this.
Reviewer 3 Report
The authors present a review of the surgical management in retroperitoneal soft tissue sarcoma. In addition to discussing the technical aspects of operative conduct, they focus on disease biology and its interplay with resection. The salient controversies are highlighted nicely and very reasonable, data-driven stances are taken when appropriate. The discussion surrounding each of these controversies is nuanced and expertly written. Attention is paid to not only the landmark trials in the RPS space, but very important smaller studies which have attempted to address surgical controversies that have asked challenging questions. The manuscript truly was a pleasure to read and avoids (appropriately so) taking a strong stance on particular issues where little data guides practice (ex. compartmental resection for RPS). The reader's attention is always redirected back to the underlying biology and how histology guides these decisions in a patient-specific manner. I have no major comments/criticisms and think this work is a strong contribution to our field.
Author Response
We thank the reviewer for their comments.
Reviewer 4 Report
There are some minor comments.
It would be better to describe what are new in this manuscript compared to previous review articels about surgical management of retroperitoneal sarcomas.
It would be better to add a determination of resection (patient selection) for surgical management.
Please modify the references according to the format of the Journal.
Author Response
We thank the reviewer for their comments. We agree highlighting what sets this review apart is beneficial for the manuscript. We have added a sentence to the end of the introduction to discuss what is unique to this review on lines 41-44.
We agree that determination of resection is a critical topic of discussion. Given that patient selection in RPS is so closely tied to achieving an R0 resection, we elected to discuss selection within the context of surgical approach.
The references have been modified according the format of the journal.